# A zeolitic vanadotungstate family with structural diversity and ultrahigh porosity for catalysis

Zhenxin Zhang[1,2], Qianqian Zhu[1], Masahiro Sadakane[3], Toru Murayama 🄳 [4], Norihito Hiyoshi[5], Akira Yamamoto[6,7], Shinichi Hata[4], Hisao Yoshida[6,7], Satoshi Ishikawa[1], Michikazu Hara[2] & Wataru Ueda[1]

Design of the structure and composition of crystalline microporous inorganic oxides is of great importance in catalysis. Developing new zeolites is one approach towards this design because of the tunable pore system and high thermal stability. Zeolites are limited to main group elements, which limits their applications in redox catalysis. Another promising choice is zeolitic transition metal oxides providing both porosity and redox activity, thereby further expanding the diversity of porous materials. However, the examples of zeolitic transition metal oxides are rare. Here, we report a new class of zeolitic vanadotungstates with tunable frameworks exhibiting a large porosity and redox activity. The assembly of $[W_4O_{16}]^{8-}$ units with $VO^{2+}$ forms two isomeric porous frameworks. Owing to the complex redox properties and open porosity, the vanadotungstates efficiently catalyse the selective reduction of NO by $NH_3$. This finding provides an opportunity for design and synthesis of inorganic multi-functional materials for future catalytic applications.

[1] Faculty of Engineering, Kanagawa University, Rokkakubashi, Kanagawa-ku, Yokohama-shi, Kanagawa 221-8686, Japan. [2] Materials and Structures Laboratory, Tokyo Institute of Technology, Nagatsuta-cho 4259, Midori-ku, Yokohama-city, Kanagawa 226-8503, Japan. [3] Department of Applied Chemistry, Graduate School of Engineering, Hiroshima University, 1-4-1 Kagamiyama, Higashi Hiroshima 739-8527, Japan. [4] Department of Applied Chemistry, Graduate School of Urban Environmental Sciences, Tokyo Metropolitan University, 1-1 Minami-osawa, Hachioji, Tokyo 192-0397, Japan. [5] Research Institute for Chemical Process Technology, National Institute of Advanced Industrial Science and Technology (AIST) 4-2-1 Nigatake, Miyagino, Sendai 983-8551, Japan. [6] Graduate School of Human and Environmental Studies, Kyoto University, Yoshida-nihonmatsu-cho, Sakyo-ku, Kyoto 606-8501, Japan. [7] Elements Strategy Initiative for Catalysts & Batteries (ESICB), Kyoto University, Kyotodaigaku Katsura, Nishikyo, Kyoto 615-8520, Japan. Correspondence and requests for materials should be addressed to Z.Z. (email: pd196277hd@kanagawa-u.ac.jp) or to W.U. (email: uedaw@kanagawa-u.ac.jp)

Zeolites, microporous crystalline aluminosilicates, are inorganic functional materials that have wide applications in catalysis, because of their well-defined structures, complex pore systems, thermal stabilities, and high surface areas. To date, hundreds of zeolitic frameworks with a variety of ordered microporous topological structures have been discovered[1–5]. On the other hand, the composition of zeolites is still limited to a few main group elements, such as Si, Al, P, B, and Ge, which hinders the further development of zeolites.

The combination of properties derived from the composition with the specific porous structure is more important for the future development and application of zeolitic materials in catalysis. Transition metal elements have superior properties including redox, photochemical, electrochemical, and magnetic properties due to partially occupied d-orbitals, which enable catalytic functions and applications that main group elements cannot achieve. With the combination of the unique properties of transition metals with ordered microporosity, major breakthroughs in applications have been achieved[6–8]. The classical methods to achieve this kind of combination is the direct modification of zeolites through ion-exchange[9], modifying the framework disorderly[10], and forming mixed tetrahedral and octahedral frameworks[11,12]. However, these approaches are not effective for introducing transition metals with a high content, precise locations, and well-ordered structure.

A more inventive method for forming a porous framework is the direct assembly of transition metal ions via a bottom-up approach, particularly using polyoxometalates (POMs) as cluster precursors[13]. There are only a few examples of zeolitic transition metal oxides. The typical materials are porous frameworks based on ring-shaped $[P_8W_{48}O_{184}]$ unit[14,15], $[MnV_{13}O_{38}]$ unit[16], and pentagon $[Mo_6O_{21}]$ units[17]. However, the current materials suffer from problems that include poor structural diversity, unopened micropores[18], and low surface area and porosity[18,19]. Therefore, new progresses in obtaining zeolitic materials with all-transition metal oxides that would increase the diversity of material design are strongly desired by material scientists.

Here, we report the synthesis of a new family of zeolitic transition metal oxides based on vanadotungstate (**VT**), which successfully combine a transition metal composition and porosity with high structural and property tunability. The **VT** family is constructed by connecting cubane clusters, $[W_4O_{16}]^{8-}$, with $VO_2^+$ linkers to generate various isomeric frameworks. The structural prediction reveals the two most stable frameworks, which are able to be synthesized accordingly. The structures of the new materials are confirmed by powder X-ray diffraction (XRD), high-angle annular dark-field scanning transmission electron microscopy (HAADF-STEM), and X-ray absorption near-edge structure (XANES) analysis, showing that different arrangements of the units and linkers form porous frameworks with LTA and IRY topologies, respectively. The micropores are opened, and the materials adsorb a variety of small molecules. The redox-active transition metal oxide porous framework exhibits high catalytic activity in the selective reduction of NO by $NH_3$.

## Results

### Synthesis and characterization of VT-1.
The vanadotungstate was synthesized by a hydrothermal method in a weakly acidic precursor solution of tungstate and vanadyl sulphate, which was denoted as **VT-1**. Vanadyl sulphate was essential to obtain the material (Supplementary Fig. 1 and Table 1), and the material did not form using other V compounds. The XRD pattern and Fourier transform infrared (FTIR) spectrum of this material are shown in Fig. 1. **VT-1** was a polyhedral-shaped material with a diameter of ca. 100 nm (Supplementary Fig. 2a), which was not suitable for single-crystal X-ray analysis. Alternatively, powder XRD, HAADF-STEM observations, and XANES analysis were used for structural determination.

The XRD pattern of **VT-1** was indexed to a cubic cell with lattice parameters of 17.1101 Å and a space group of PA-3 (Supplementary Table 2). The initial structure was solved by the charge-flipping algorithm (Supplementary Table 3). The arrangement of W formed $[W_4O_{16}]^{8-}$ cubane cluster[20]. The $[W_4O_{16}]^{8-}$ units were proposed to be connected by $VO^{2+}$ linkers to form a 3D framework. The simulated pattern fit well to the experimental pattern after Rietveld refinement, indicating that the proposed structure was correct and that there were no obvious crystalline impurities (Supplementary Fig. 3a and Table 4).

The arrangement of the $[W_4O_{16}]^{8-}$ units in **VT-1** was visualized by HAADF-STEM (Fig. 2a). The lattice parameters determined from HAADF-STEM were the same as those determined from powder XRD. The ordered arrangement of four intense spots was observed in the (100) plane, which was attributed to the $[W_4O_{16}]^{8-}$ units. The periodic packing of the $[W_4O_{16}]^{8-}$ units in the (100) and (110) planes was identical to the proposed structure (Fig. 2a, b).

The W $L_1$-edge positions of **VT-1** were similar to those of $Ba_2NiW^{VI}O_6$, $W^{VI}O_3$, and $Na_2W^{VI}O_4$ (Supplementary Fig. 4), indicating that $W^{VI}$ was present in **VT-1**. The pre-edge peak intensity of **VT-1** was similar to that of $WO_3$, which is composed of distorted octahedral $WO_6$ units[21]. Tetrahedral $WO_4$ units show a larger pre-edge intensity than octahedral $WO_6$ units[21]. Therefore, **VT-1** was based on distorted octahedral $WO_6$ units. For the V–K edge, the edge position of **VT-1** was similar to that of $VO_2$, indicating that $V^{IV}$ was present in the material. The pre-edge

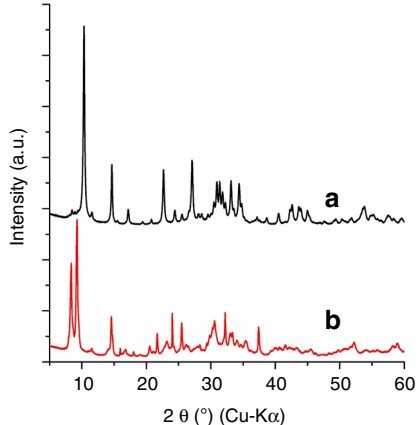

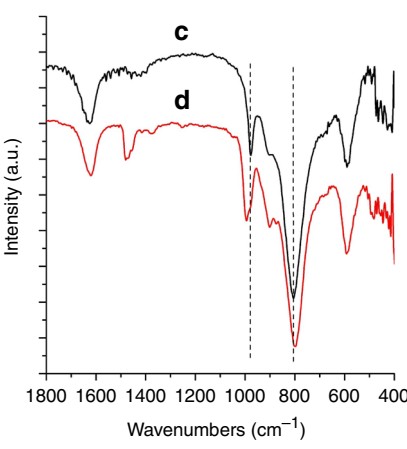

**Fig. 1** XRD patterns and FTIR spectra of the materials. XRD patterns of **a VT-1** and **b VT-5**, FTIR spectra of **c VT-1** and **d VT-5**

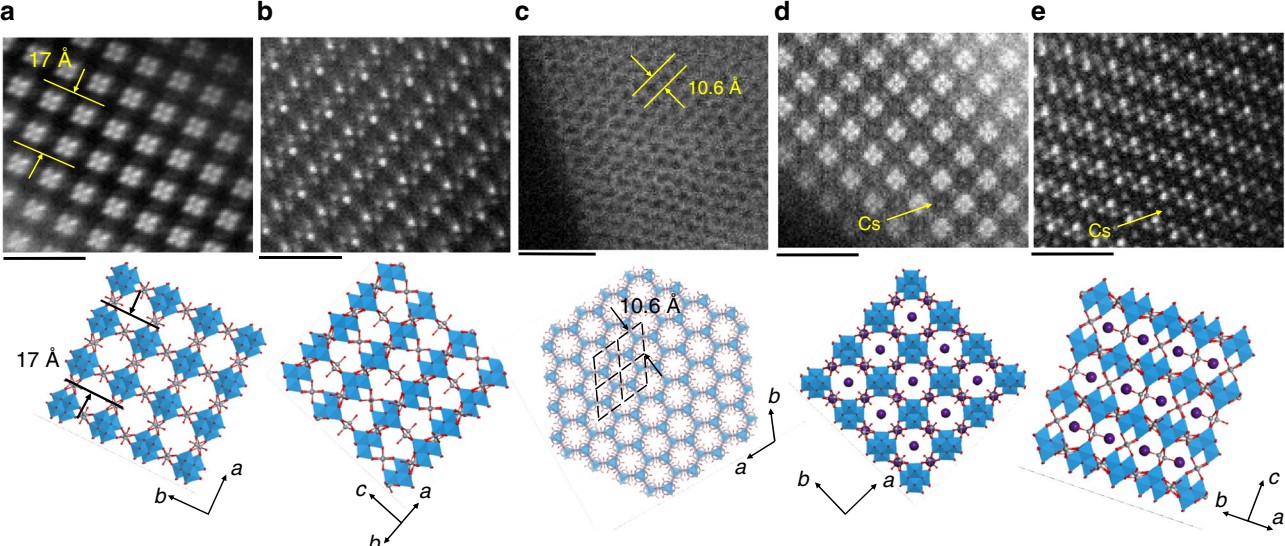

**Fig. 2** HAADF-STEM images (top) and corresponding structure models (bottom). **a VT-1** in the (1 0 0) plane, scale bar: 2 nm, **b VT-1** in the (1 1 0) plane, scale bar: 2 nm, **c VT-5** in the (0 0 1) plane, scale bar: 5 nm, **d Cs-VT-1** in the (1 0 0) plane, scale bar: 2 nm, and **e Cs-VT-1** in the (1 1 0) plane, scale bar: 2 nm; W (blue), V (grey), O (red), and Cs (purple)

peak is affected by the valence state and coordination environment of the central atom[22,23]. The pre-edge peak intensity of **VT-1** was comparable to that of $\alpha$-VOSO$_4$[24]. The coordination environment of V was similar to that in $\alpha$-VOSO$_4$ where V is coordinated by six oxygens in an octahedral fashion and is displaced towards an oxygen to give a short V–O distance characteristic of VO$^{2+}$.

X-ray photoelectron spectroscopy (XPS) (Supplementary Fig. 5a,b) and bond valence sum (BVS) calculation (Supplementary Table 5) further confirmed the presence of W$^{VI}$ and V$^{IV}$. The reduced V species were able to be confirmed by XPS, which was similar to the POM system[25]. O (1s) showed two different simulated peaks, indicating the existence of a framework O with low binding energy and O from surface adsorbates with high binding energy (Supplementary Fig. 5c)[25]. The diffuse reflectance ultraviolet visible (DR-UV-vis) spectra indicated that the reduced V species showed signals at ca. 820 and 645 nm (Supplementary Fig. 6a). Elemental analysis demonstrated that the ratio of K: N: W: V: H = 1.5: 0.25: 4: 3: 20.25 for **VT-1**. Therefore, the chemical formula of **VT-1** was estimated to be (NH$_4$)$_{0.25}$K$_{1.5}$H$_{0.25}$[W$^{VI}_4$V$^{IV}_3$O$_{19}$]·9.5H$_2$O.

The units of **VT-1**, [W$_4$O$_{16}$]$^{8-}$ comprised four tetrahedrally linked WO$_6$ octahedra and were connected to six nearby VO$^{2+}$ linkers in an octahedral fashion through coordination to the terminal W=O bonds of the units (Fig. 3a). The micropore opening was formed by an eight-membered oxygen ring and had a diameter of ca. 4.3 × 4.3 Å (Fig. 3b). The channel was connected and unblocked, forming a 3D pore system in the material (Fig. 3c).

Inorganic zeolitic materials are classified into three groups based on transition metal component (Supplementary Table 6). First group is zeolites composed of main group elements. The second group is tetrahedral–octahedral frameworks[26], which orderly incorporate transition metal oxygen octahedra in the tetrahedral silicoaluminate frameworks, such as Pharmacosiderite minerals[27] and Tschörtnerite minerals[28]. The third group is zeolitic transition metal oxides, the porous frameworks of which are composed of fully transition metal oxides. **VT-1**, as a typical zeolitic transition metal oxide, had the same structural topology as LTA zeolites and Pharmacosiderite minerals[27]. LTA zeolites are based on SiO$_4$ tetrahedra (Supplementary Fig. 7a, b), and

Pharmacosiderite minerals are composed of [Ti$_4$O$_{16}$]$^{16-}$ units with tetrahedral Si$^{4+}$ linkers, which are different from **VT-1**. Tschörtnerite is another related material[28]. The porous framework is constructed by connection of SiO$_4$ and AlO$_4$ with the openings of 4.2 × 4.2 and 5.6 × 3.1 Å, respectively. The [Cu$_{12}$O$_{24}$] cluster is trapped in the micropore, which is a POM cluster[29], and there is only Van der Waals interaction between the framework and [Cu$_{12}$O$_{24}$]. The comparison of the related materials demonstrated that **VT-1** represented unique zeolitic transition metal oxide, which combine the fully transition metal oxide framework with ordered microporosity.

**Structural evolution**. Compared with organic frameworks and zeolites, the structural diversity of transition metal oxides is considered to be poor. However, connecting [W$_4$O$_{16}$]$^{8-}$ units with VO$^{2+}$ linkers could generate a variety of structures. To generate isomeric structures based on [W$_4$O$_{16}$]$^{8-}$ and VO$^{2+}$ (Supplementary Fig. 8), the connection between [W$_4$O$_{16}$]$^{8-}$ and the linker was restricted (See Methods, Structural evolution and prediction). There were two reasonable types of linkages: one where VO$^{2+}$ binds to two terminal oxygens of two neighbouring WO$_6$ (type i), and one where VO$^{2+}$ binds to two terminal oxygens of one WO$_6$ (type ii) (Supplementary Fig. 9). All the possible connections of the units and linkers are listed in Supplementary Table 7, and among these connections only three types were reasonable; these types are denoted the preliminary building units (A, B, and C in Fig. 4a). Two preliminary building units further connected with each other to form secondary building units (D–H in Fig. 4a). In total, eight building units were obtained, and these building units constructed ten hypothetical structures, denoted as **VT-1** to **VT-10**, respectively (Fig. 4b and Supplementary Table 8). The XRD patterns of the hypothetic structures were calculated (Supplementary Fig. 10), and the corresponding crystallographic data are listed in Supplementary Table 8- Table 18. It is interesting that the **VT** family had a higher porosity (see Methods, Computer-based calculations and simulations for detailed calculation) than zeolites (Supplementary Table 19), and most of them had over 50% porosity, which was higher than 90% of zeolites, including some widely used zeolites, such as FAU (46.86%), LTA (43.79%), MWW (42.05%), MOR (38.23%), and

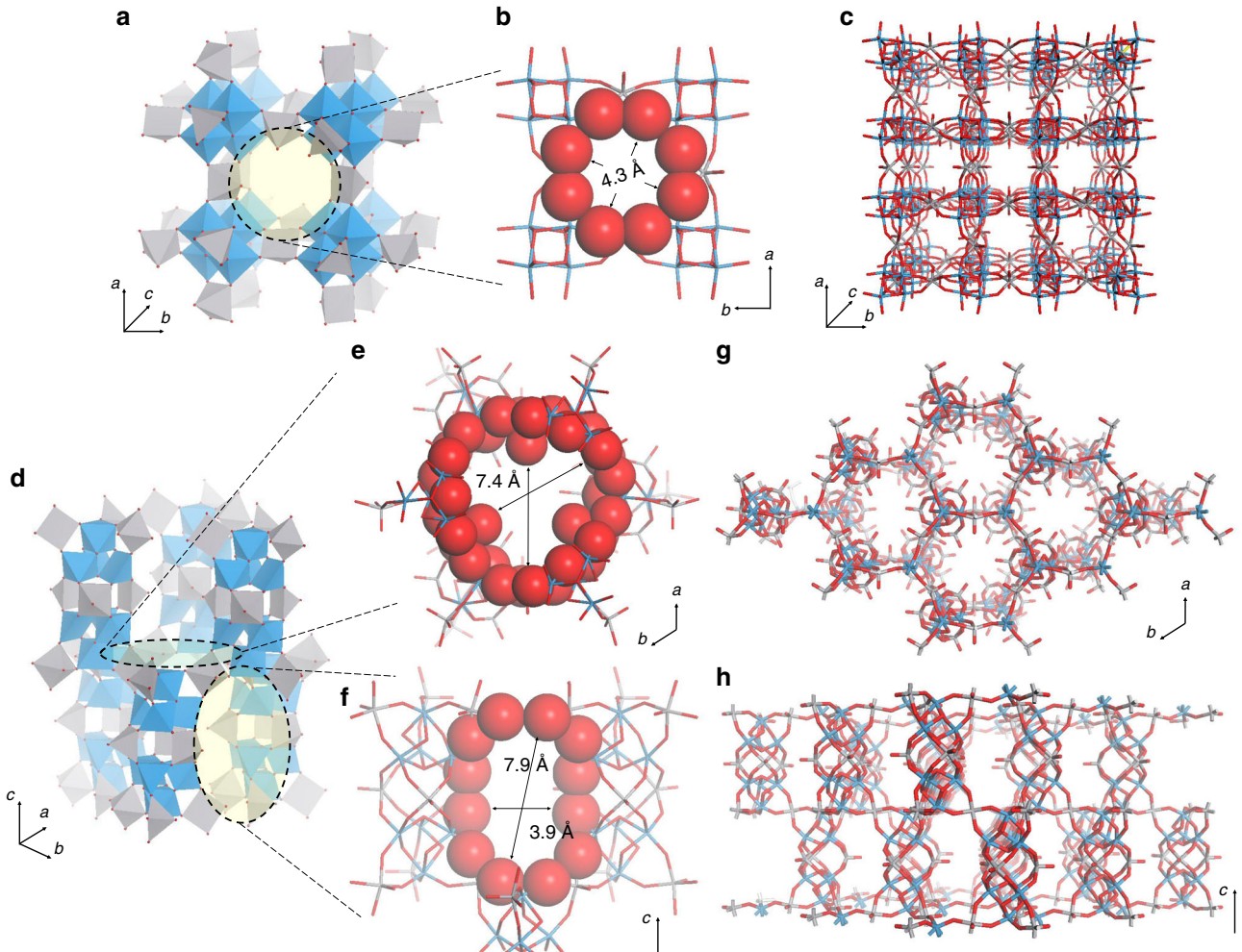

**Fig. 3** Structural models of the synthesized **VT** family. **a** Polyhedral representation of the connection of $[W_4O_{16}]^{8-}$ units with $VO^{2+}$ linkers for **VT**-1; **b** opening to the channel in **VT**-1; **c** extended framework of **VT**-1; **d** polyhedral representation of the connection of $[W_4O_{16}]^{8-}$ units with $VO^{2+}$ linkers for **VT**-5; openings to **e** the vertical channel and **f** the parallel channel in **VT**-5, **g** extended framework of **VT**-5 in the $a$–$b$ plane; and **h** extended framework of **VT**-5 along the $c$-axis; W (blue), V (grey), and O (red)

MFI (31.34%). There are also molecular type **VTs**, such as V (dien))$_4$[W$_2$O$_{14}$] and V(dien))$_4$[W$_4$O$_{20}$] (dien = diethylene-triamine, $C_4H_{13}N_3$)[30], and V(dien))$_4$[W$_4$O$_{20}$] is also composed of cubane unit and V ions. Formation energies ($\Delta H_f$) of the zeolitic **VTs** are lower than those of the molecular type **VTs** (Supplementary Fig. 11). This indicated that the zeolitic **VTs** might be more stable.

The chemical compositions ($[W_4V_3O_{19}]$) of the hypothetical frameworks were the same, while the structures were different. The structures were optimized by density functional theory (DFT) calculations, and the energies and densities of the structures were obtained and plotted, which is useful in structural predictions[31]. **VT**-1 showed the lowest energy and highest density among all the **VT** members, indicating that **VT**-1 was the thermodynamically most stable structure (Fig. 4c). Furthermore, for the rest of the hypothetical frameworks, the density of the frameworks was lower than that of **VT**-1, indicating that the porosity might be higher (Fig. 4c). The energies of the frameworks in **VT**-2 to **VT**-10 were close to that of **VT**-1, indicating that they were also good candidates. In particular, the energy of **VT**-5 was just 1.1 kJ mol$^{-1}$ higher than that of **VT**-1, and therefore, this structure might be the next most suitable target structure for synthesis. The calculation results were only based on

the bare frameworks, and the structures might be further stabilized after the incorporation of suitable cation species.

**Synthesis and characterization of VT-5.** The computer-based simulation suggested the existence of **VT**-5. Indeed, encouraged by the simulation results, this structure was successfully synthesized. The cation played a key role in directing the formation of the different phases of **VT**. K$^+$ led to the formation of **VT**-1 and trimethyl ammonium cation (TMA) cation led to the formation of the less dense **VT**-5.

The XRD patterns of **VT**-1 and **VT**-5 demonstrated that the crystalline phase of these two materials were different (Fig. 1a, b). Interestingly, the IR spectra of the frameworks of **VT**-1 and **VT**-5 were similar, which indicated the presence of the same unit ($[W_4O_{16}]^{8-}$) and linkage mode in both materials (Fig. 1c, d). A slight IR band shift was observed, indicated that the bonding state of **VT**-5 was slightly different from that of **VT**-1. For **VT**-5, a peak at 1480 cm$^{-1}$ was observed for TMA. The W–L edge and the V–K edge XANES peaks of **VT**-5 were almost the same as those of **VT**-1, which demonstrated the presence of distorted $W^{VI}O_6$ for W and $V^{IV}O^{2+}$ for the linker ions (Supplementary Fig. 4). XPS (Supplementary Fig. 5d–f), DR-UV-vis

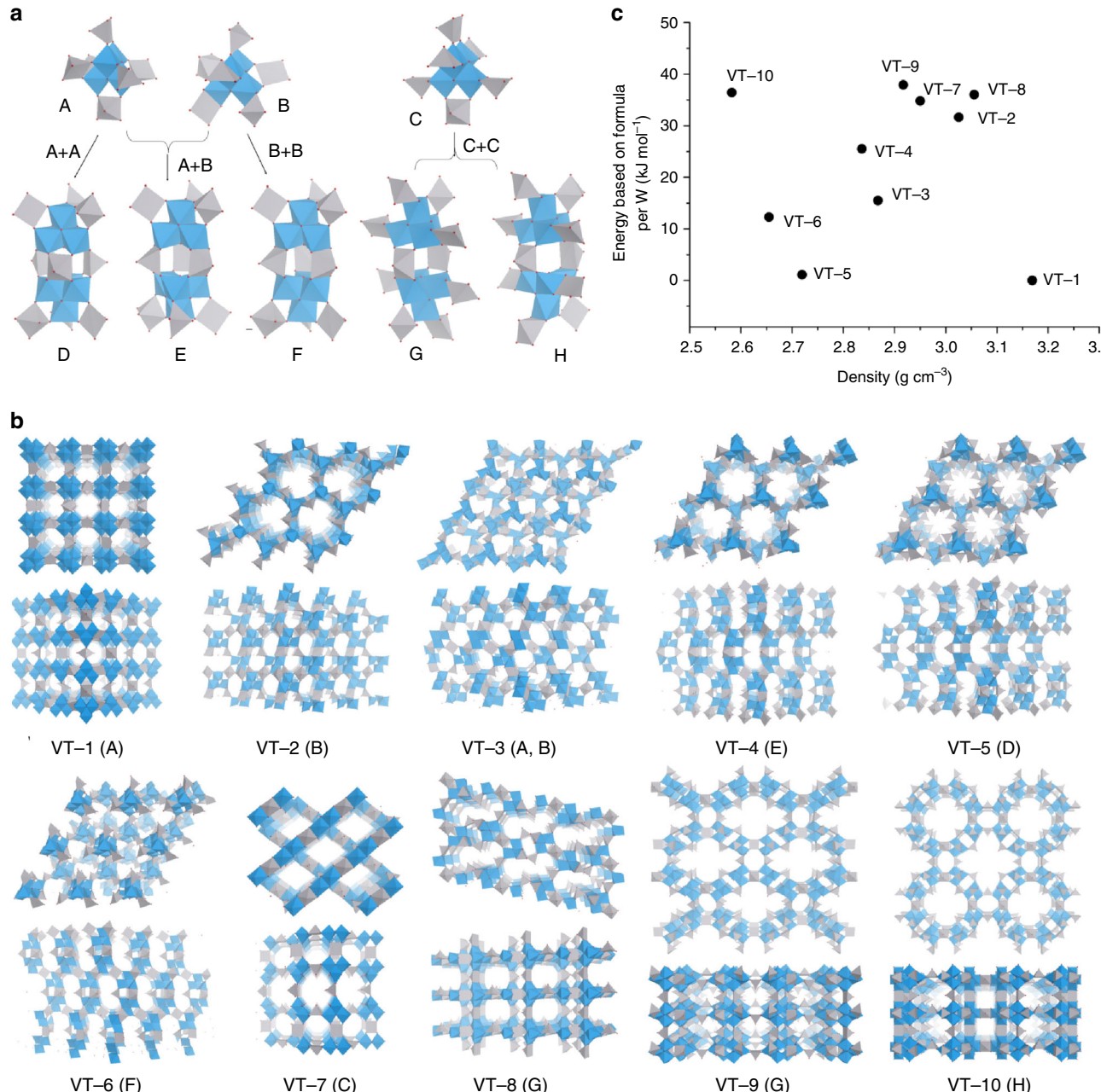

**Fig. 4** Models of the building units and corresponding hypothetical structures. **a** Three preliminary building units (top) and five secondary building units (bottom); **b** frameworks of the VT zeolite family constructed by the units from the front view (top) and side view (bottom); W (blue), V (grey), O (red); and **c** energy–density map for the proposed structures

(Supplementary Fig. 6b), and BVS (Supplementary Table 20) combined with elemental analysis demonstrated that the chemical formula of **VT-5** was $[[N(CH_3)_3]_{1.4}H_2[W^{VI}_4V^{IV}_3O_{19}]\cdot 2.5H_2O$. Thus, **VT-1** and **VT-5** were isomeric materials.

**VT-5** formed thin-plate crystals (Supplementary Fig. 2b). Indexing showed that the unit cell, lattice parameters, and space group were in good agreement with the hypothetical structure of **VT-5** (Supplementary Table 2 and Table 8). Rietveld refinement confirmed the structure of **VT-5**, and the experimental pattern was identical to the simulated one (Supplementary Fig. 3b and Table 21). The HAADF-STEM image of **VT-5** in the $a$–$b$ plane exhibited a hexagonal pattern, and the $d$-spacing of the (100) plane obtained from the TEM image was almost the same as that

of the proposed structure. Channels vertical to the $a$–$b$ plane were observed, and these channels had a hexagonal arrangement (Fig. 2c).

Two $[W_4O_{16}]^{8-}$ units were connected to each other via three $VO^{2+}$ linkers, forming a secondary building unit with trigonal symmetry. Six $VO^{2+}$ linkers further linked the secondary building units to form a porous network. There were two different types of channels in the material (Fig. 3d). The channel that was vertical to the $a$–$b$ plane was observed to have trigonal symmetry, the opening to which had a size of ca. 7.4 × 7.4 Å and was surrounded by oxygens (Fig. 3e). The other channel was parallel to the $a$–$b$ plane, and the opening was composed of 12 oxygen atoms with a prolate shape and a size of ca. 3.9 × 7.9 Å

(Fig. 3f). The vertical channels were cross-linked by parallel channels to form a 3D microporous structure in **VT-5**, which was completely different from the structure of **VT-1**. The structure of **VT-5** was the same as that of the zeolite IRY (Supplementary Fig. 7c, d).

**Cation adsorption properties**. **VT-1** exhibited ion-exchange properties. The original cations, $K^+$ and $NH_4^+$, were completely replaced by $Cs^+$ without changing the structure based on XRD, FTIR, and elemental analysis (Supplementary Fig. 12), and the resulting material was denoted **Cs-VT-1**. The $Cs^+$ position in **Cs--VT-1** was investigated by HAADF-STEM. $Cs^+$ was more facilitate to be observed by TEM. The HAADF-STEM image of **Cs-VT-1** showed that the arrangement of the $[W_4O_{16}]^{8-}$ units was the same as that of **VT-1** in the (100) and (110) planes (Fig. 2d, e). A new signal was observed in the centre of the four $[W_4O_{16}]^{8-}$ units, which was attributed to $Cs^+$. According to the HAADF-STEM images, $Cs^+$ was located in the centre of the pore opening. The structure was determined by powder XRD (Supplementary Table 22) and further refined by the Rietveld method, which provided results identical to those from the TEM observation (Supplementary Fig. 3c and Table 23).

**Molecule adsorption properties**. Thermogravimetric-differential thermal analysis (TG-DTA) and temperature-programmed desorption mass spectrometry (TPD-MS) (Supplementary Fig. 13) demonstrated that the occupied water could be removed from **VT-1** and **VT-5** at 100–200 °C. For **VT-5**, the second weight loss observed by TG-DTA was attributed to the decomposition of TMA because $CO_2$ and $NH_3$ were detected by TPD-MS. The materials were stable up to 300 °C in $N_2$ without changes in the XRD and FTIR results (Supplementary Figs. 14, 15). Therefore, water was removed by calcination without decomposing the structures of both materials. The decomposition temperature of **VT-5** (300 °C) was lower than that of TMA (Supplementary Fig. 13d), which made the removal of TMA by calcination difficult for **VT-5**.

The porosity of **VT-1** was analysed through $N_2$ adsorption-desorption measurements after the removal of water. The results showed a type-I adsorption isotherm, demonstrating that the material was microporous (Fig. 5a). The surface area and micropore volume were 310 m$^2$ g$^{-1}$ and 0.102 cm$^3$ g$^{-1}$, respectively. The pore-size distribution was narrow with a diameter of 0.49 nm, demonstrating that the material had ordered intrinsic micropores and that only a single-channel system existed (Fig. 5b). $K^+$ in the material did not completely block the micropores, because the number of $K^+$ (1.5 per a formula) was not sufficient. Microporosity and surface area of the material decreased dramatically after the structure was damaged (Supplementary Fig. 16).

**VT-1** adsorbed a variety of small molecules. The normal alkane molecules, $CH_4$, $C_2H_6$, $C_3H_8$, and n-$C_4H_{10}$, were adsorbed (Fig. 5c). **VT-1** did not adsorb alkanes with side branches or cycloalkanes, because the size of those molecules was larger than the size of the pore opening. The micropore diameter of **VT-1** was determined by plotting the micropore volume estimated by the Dubinin–Astakhov (DA) method against the kinetic diameter of each molecule to give a diameter between 0.43 and 0.5 nm, which was in good agreement with the results from crystal structure analysis and $N_2$ adsorption measurements (Fig. 5d). Polar molecules, such as water, $CO_2$, acetone, methanol, and ethanol, could also be adsorbed by **VT-1** (Fig. 5e). The ratio of adsorbed molecules per cubane unit was above one in most cases (Supplementary Table 24), indicating that adsorption occurred in the micropores. Thus, **VT-1** has potential for application in gas separation.

Compared with other frameworks based on transition metal oxides, **VT-1** had a higher porosity and surface area (Supplementary Table 25). Even compared with zeolites (Supplementary Table 25)[32–36], **VT-1** had a comparable surface area based on weight even though **VT-1** was composed of fully transition metal elements, which are heavier than Si.

The unremovable organic compounds in the micropores of **VT-5** caused the adsorption properties of **VT-5** to be different from those of **VT-1**. **VT-5** did not adsorb $N_2$, although the porosity of **VT-5** was larger than that of **VT-1** in the crystal structures (Supplementary Fig. 17). However, the pores of **VT-5** were still accessible to other small molecules (Fig. 5f). The adsorbed amount of **VT-5** was lower than that of **VT-1**, and the number of adsorbed molecules was listed in Supplementary Table 24.

The structures of **VT-1** and **VT-5** were stable during the adsorption experiments. After adsorption, the used materials were characterized by XRD, which indicated that the structure did not change (Supplementary Fig. 18).

**Catalytic activity**. Selective catalytic reduction of NO with $NH_3$ ($NH_3$-SCR) is widely used for the removal of the NO pollutant produced by automobiles and power plants[37]. The reaction requires catalysts with acidity, redox properties, and high surface area. There are efficient SCR catalysts being reported, such as V-based catalysts[38–43], Mn-based catalysts[44], and Cu-zeolite catalysts[45]. The present supported catalysts were composed of porous supports and redox active transition metals (or oxides) with the classical design concept of simple mixing of porosity and redox properties[46,47]. The zeolitic **VT** represented another catalyst design concept that simultaneously achieved an order complex microporosity, acidity, and redox properties in a single crystal, which would offer new opportunities for improving the activity and indicated the potential of the zeolitic transition metal oxides in catalysis.

$NH_3$-SCR was used as a model reaction to test the activity of **VT-1**. **VT-1** acted as an efficient catalyst for $NH_3$-SCR. As shown in Fig. 6, the activity of **VT-1** continually increased with increasing temperature. When the temperature reached 120 °C, the conversion of NO reached 82% with 100% of $N_2$ selectivity. The structure was remained after reaction at 120 °C, which indicated that the material was stable enough to achieve high activity without structural collapse (Supplementary Fig. 19a, b). The further increase of the temperature to 200 °C decreased the activity, which corresponded to the deformation of structure at high temperature (Supplementary Fig. 19c).

The activity of **VT-1** was strikingly higher than the single metal oxides, $V_2O_5$, $VO_2$, and $WO_3$ (Fig. 6a). Furthermore, the widely used conventional, $V_2O_5/TiO_2$[39–41] and $V_2O_5$-$WO_3/TiO_2$[38,42,48], were synthesized for comparison. NO conversion of **VT-1** (82%) was higher than that of $V_2O_5/TiO_2$ (30%) and $V_2O_5$-$WO_3/TiO_2$ (28%) at 120 °C. The results demonstrated that **VT-1** was more active at a low temperature (Fig. 6b), although the chemical composition of them were similar, which indicated the power of combining redox properties and a microporous structure for enhancing the catalytic activity.

Acidity[49], redox property[50], and surface area are important for achieving high catalytic activity for $NH_3$-SCR. $NH_3$-TPD exhibited that **VT-1** adsorbed $NH_3$ at 100 °C, indicating that the material was acidic (Supplementary Fig. 20). **VT-1** incorporated transition metal ions that were considered to be redox active. The material was stable after calcined at 120 °C, and XPS showed V was partially oxidized to $V^V$ (Supplementary Fig. 21), which might be the reason that $NH_3$-SCR could be catalysed at a low temperature. Furthermore, **VT-1** had high microporosity and

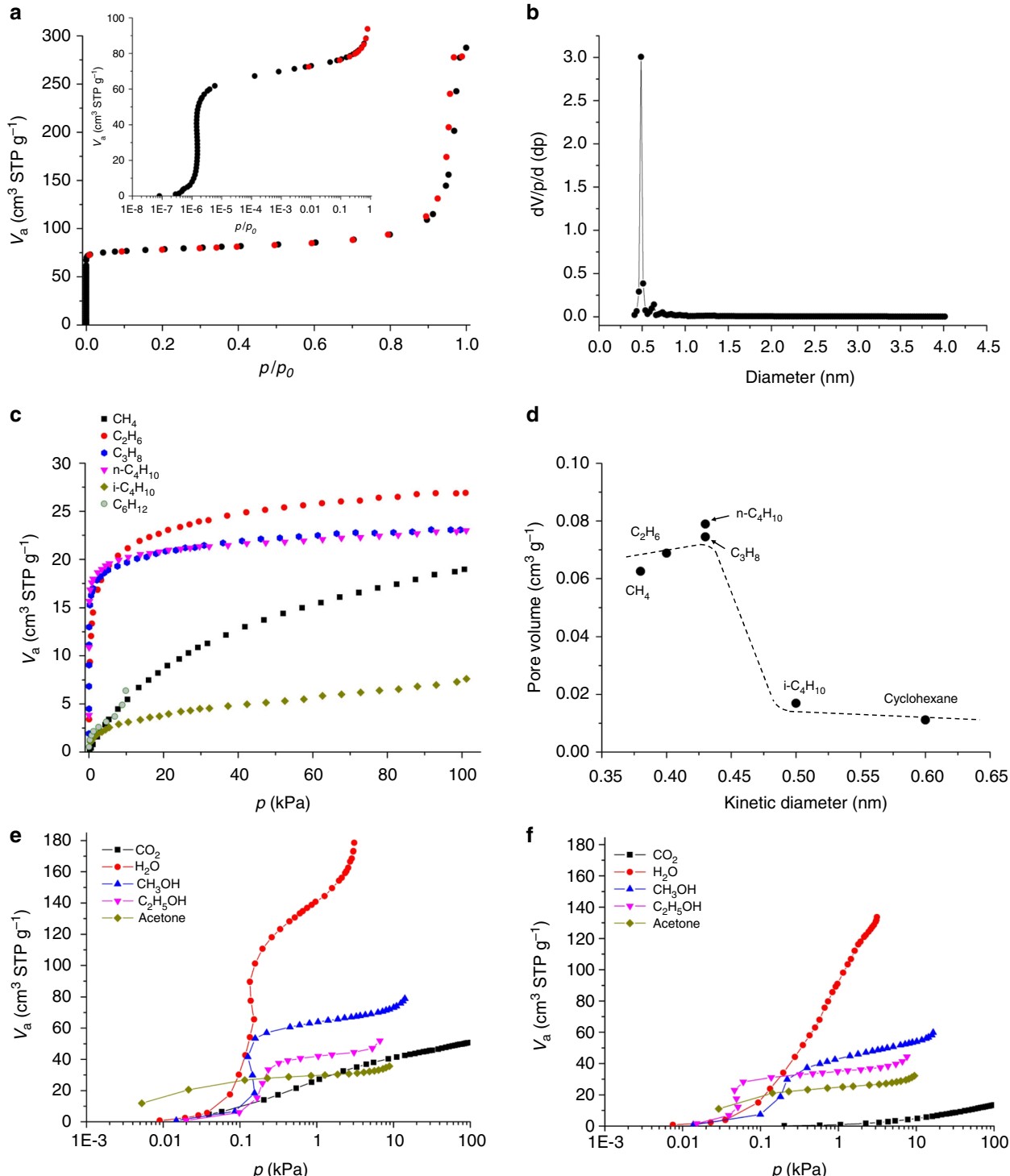

**Fig. 5** Molecular adsorption results for the materials. **a** $N_2$ adsorption-desorption isotherm of **VT**-1 at −196 °C (inset image: low pressure), adsorption (black) and desorption (red); **b** pore size distribution determined using the SF method; **c** adsorption isotherms of alkanes for **VT**-1 at 25 °C, **d** micropore volumes of **VT**-1, which were estimated by the DA method, plotted against kinetic diameter; and adsorption isotherms of small molecules for **e VT**-1 and **f VT**-5 at 25 °C

surface area, which made the acid and redox sites accessible to reactants and improved the activity further.

For practical application, water and $SO_2$ effects on the activity of the catalysts are important. We carried out $NH_3$-SCR with water. When water was introduced the activity decreased (Supplementary Fig. 22). Due to the microporosity of **VT**-1 and low reaction temperature, water might be adsorbed in the pore and occupy the

active sites, which inhibited the reaction. The Kelvin equation estimated that water preferentially adsorbed in **VT**-1 compared with $NH_3$ (Supplementary Table 26, Table 27, and discussion). Furthermore, **VT**-1 would react with $SO_2$ at 130 °C, the activity of which might be also inhibited by $SO_2$ (Supplementary Fig. 23). Therefore, there problems that would limit the practical applications of the material should be addressed in the future.

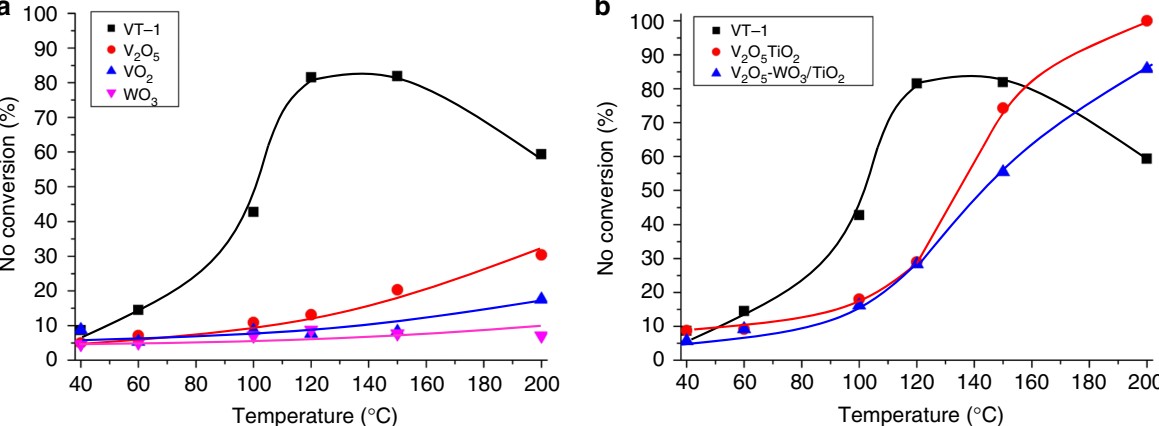

**Fig. 6** Comparison of catalytic activity of **VT-1** and other catalysts for $NH_3$-SCR. **a** Single-metal oxides and **b** mixed-metal oxides. Reaction conditions: catalyst: 0.15 g with the secondary particle size of 38–106 μm, total flow rate: 250 mL min$^{-1}$, NO: 250 ppm, $NH_3$: 250 ppm, $O_2$: 4 vol% (Ar as the diluter)

## Discussion

We developed a new catalogue of catalytic material that combined high porosity and redox properties in a single crystalline vana-dotungstate, which was composed of $[W_4O_{16}]^{8-}$ units with $VO^{2+}$. The structures of the newly synthesized materials were carefully confirmed by HAADF-STEM, X-ray diffraction, XANES, and other techniques. The different assembly styles of the $[W_4O_{16}]^{8-}$ units produced cubic **VT-1** and trigonal **VT-5**. The micropores of the **VT** materials could be opened without collapse of the basic structures and were accessible to small molecules. **VT-1** was used as an efficient catalyst for $NH_3$-SCR. The catalytic performance of **VT-1** at a high reaction temperature and in the presence of water and $SO_2$ would be improved in the view of practical applications, but the material was more active than those employed with traditional V-based catalysts at a low temperature under an ideal reactant composition (dry gas without $SO_2$).

Theoretical calculations provided a promising approach towards predicting the structures and led to the synthesis of new zeolitic transition metal oxides. The assembly of the cubane unit with an inorganic linker had a variety of structural possibilities, which expanded zeolitic materials to include fully transition metal oxides. Moreover, transition metal oxide has high elemental diversity, and the elemental composition of the cubane cluster and linker may be tuned, which enables the formation of a number of iso-structural materials[20,51,52]. Therefore, this research offered an opportunity to design and develop multifunctional microporous transition metal oxides with high structural and elemental diversity, tunability, and complexity for future catalysis.

## Methods

**Material synthesis**. Synthesis of **VT-1**. $WO_3$ (4.637 g, 20 mmol) was dissolved in 200 mL of KOH (85%) (3.634 g, 55 mmol) aqueous solution at 100 °C. After the solution cooled to room temperature, it was acidified with 22 g of $H_2SO_4$ (1 M). VOSO$_4$ (2.678 g, 11 mmol) was added to the solution followed by stirring for 10 min. Finally, 0.6 mL of $NH_3$ (28%) was added to the solution, and the pH was ca. 4. The mixture was transferred to glass tubes that were sealed in a 300-mL Teflon-lined stainless-steel autoclave, which was heated at 175 °C for 8 h. After the hydrothermal reaction, the crude solid was obtained by centrifugation (2330 × g, 5 min). The crude solid was dispersed in 120 mL of water, and the mixture was centrifuged (190 × g, 2 min). The upper solution (ca. 70%) was collected. This process was repeated three times. All of the collected solutions were centrifuged (190 × g, 2 min) again, and 70% of the upper solution was recovered, in which the pure sample was collected by centrifugation (2330 xg, 30 min). The resulting solid was dried at 60 °C for 24 h. Then, 1.3 g of **VT-1** was obtained (yield of 18% based on W). Elemental Analysis: Calcd for $K_{1.5}N_{0.25}W_4V_3O_{28.5}H_{20.25}$: K, 4.11; N, 0.25; W, 51.55; V, 10.71; H, 1.42; S, 0, Found: K, 4.24; N, 0.25; W, 51.72; V, 10.43; H, 1.42; S, 0.02.

Synthesis of **VT-5**. $WO_3$ (4.637 g, 20 mmol) was dissolved in a 30% trimethylamine (TMA) aqueous solution (6.6 g, 33.6 mmol) with 30 mL of water at

100 °C. After the solution cooled to room temperature, it was diluted with 170 mL of water. The solution was acidified by 11.2 g of $H_2SO_4$ (1 M). VOSO$_4$ (2.678 g, 11 mmol) was added to the solution followed by stirring for 10 min, and the pH was ca. 4. The mixture was transferred into glass tubes that were sealed in a 300-mL Teflon-lined stainless-steel autoclave, which was heated at 175 °C for 8 h. After the hydrothermal reaction, the crude solid was obtained by filtration. The crude solid was dispersed in ca. 40 mL of water, and 80% of the upper solution was collected. This process was repeated three times. All of the collected solutions were filtered and washed with water three times. The resulting solid was dried at 60 °C for 24 h. Then, 0.75 g of **VT-5** was obtained (yield of 11% based on W). Elemental Analysis: Calcd for $C_{4.2}N_{1.4}W_4V_3O_{21.5}H_{19.6}$: C,3.81; N, 1.48; W, 55.63; V, 11.56; H, 1.48, Found: C, 3.84; N, 1.52; W, 55.64; V, 12.00; H, 1.54.

For comparison, the supported V oxide catalysts were prepared using a wet impregnation method[53]. Commercial TiO$_2$ P25 (ca. 20 nm of size) was used as a support, and $NH_4VO_3$ and $(NH_4)_6[H_2W_{12}O_{40}]\cdot nH_2O$ were used as V and W precursors, respectively. For V$_2$O$_5$/TiO$_2$, $NH_4VO_3$ (0.064 g) was dissolved in water and were mixed with TiO$_2$ P25 (1 g). The mixture was stirred at room temperature for 30 min, followed by sonication for 2 h. The mixture was dried at 110 °C overnight and calcined at 500 °C for 4 h in air. For V$_2$O$_5$-WO$_3$/TiO$_2$, $NH_4VO_3$ (0.064 g) and $(NH_4)_6[H_2W_{12}O_{40}]$ (0.113 g) were dissolved in water and were mixed with TiO$_2$ P25 (1 g). The mixture was stirred at room temperature for 30 min, followed by sonication for 2 h. The mixture was dried at 110 °C overnight and calcined at 500 °C for 4 h in air.

**Characterization**. XRD patterns were obtained on an Ultima IV X-ray Diffractiometer (Rigaku, Japan) with Cu Kα radiation (tube voltage: 40 kV, tube current: 40 mA). FTIR spectroscopy was carried out on a JASCO-FT/IR-6100 instrument (JASCO, Japan). DR-UV-vis spectra were obtained using a JASCO V-550 UV-vis-spectrophotometer equipped with an ISN-470 reflectance spectroscopy accessory (JASCO, Japan). XPS was performed on a JPS-9010MC (JEOL). The spectrometer energies were calibrated using the $C_{1s}$ peak at 284.7 eV.

TPD-MS measurements were carried out from 40 °C to 500 °C at a heating rate of 10 °C min$^{-1}$ under He flow (flow rate: 50 mL min$^{-1}$). Samples were set up between two layers of quartz wool. A TPD apparatus (BEL Japan, Inc., Japan) equipped with a quadrupole mass spectrometer (M-100QA. Anelva) was used to detect $NH_3$ (m/z = 16), $H_2O$ (m/z = 18), and $CO_2$ (m/z = 44). TG-DTA was carried out up to 500 °C at a heating rate of 10 °C min$^{-1}$ under nitrogen flow (flow rate: 50 mL min$^{-1}$) on a Thermo plus TG-8120 (Rigaku, Japan). $NH_3$-TPD was conducted using the same apparatus. First, the material was calcined in He (50 mL min$^{-1}$) for 2 h at 150 °C. The $NH_3$ adsorption was carried out at 100 °C. Finally, a TPD-MS measurement was conducted to check the $NH_3$ desorption.

W L$_1$-dege and V–K edge X-ray absorption fine structure measurements were carried out in transmission mode using a Si(111) double-crystal monochromator at the Photon Factory in the High Energy Accelerator Research Organization (KEK-PF), Japan.

Reference samples of Ba$_2$NiW$^{VI}$O$_6$, WO$_3$, Na$_2$WO$_4$, V$_2$O$_3$, VO$_2$, NaVO$_3$, and V$_2$O$_5$ were commercially available. Reference samples of Ba$_2$NiWO$_6$[21] and α-VOSO$_4$[54] were prepared according to the reported procedures. Ba$_2$NiWO$_6$ was comprised of regular octahedral WO$_6$, and WO$_3$ and Na$_2$WO$_4$ were composed of distorted octahedral WO$_6$ and tetrahedral WO$_4$, respectively. The pre-edge peak intensity increases with increasing degree of distortion due to the p–d hybridization[22,23]. For the V–K edge, we measured reference samples of V$^{III}_2$O$_3$ (VO$_6$ unit), V$^{IV}$O$_2$ (VO$_6$ unit), α-V$^{IV}$OSO$_4$ (VO$_5$ unit), NaV$^V$O$_3$ (VO$_4$ unit), and V$^V_2$O$_5$ (VO$_5$ unit).

Field emission scanning electron microscopy (FE-SEM) images were obtained with an FE-SEM S-5500 system (Hitachi, Japan). HAADF-STEM images were obtained with an ARM-200F electron microscope (JEOL, Japan) operated at

200 kV with a CEOS probe aberration corrector. The probe convergence semi-angle was 14 mrad and the collection angle of the HAADF detector was 54–175 mrad. The obtained images were treated to extract periodic structures.

The elemental compositions were determined at the Material Analysis Suzukake-dai Center, Technical Department, Tokyo Institute of Technology.

**Structural determination by powder X-ray diffraction**. The initial structures of **VT-1** and **Cs-VT-1** were determined by powder XRD. The powder XRD patterns for structural analysis were obtained on a RINT2200 instrument (Rigaku, Japan) with Cu Kα radiation (tube voltage: 40 kV, tube current: 40 mA, scan speed: 1° min$^{-1}$, step: 0.01°). First, the powder XRD pattern was indexed by the DIC-VOL06[55] and X-cell programs[56]. After performing Pawley refinement, the most reasonable space group was obtained (Supplementary Table 2). Then, the Le Bail method[57] was applied for intensity extraction with the EdPCR program. The initial structure was solved by a charge-flipping algorithm[58]. The positions and types of atoms were obtained by analysing the generated electron density maps (Supplementary Table 3 and Table 22). The framework oxygen atoms and cations that could not be found by the charge-flipping algorithm were added logically based on XANES and elemental analysis. The initial structure was further refined by Rietveld analysis.

**Rietveld refinement**. The initial structures of **VT-1** and **Cs-VT-1** were determined by the charge-flipping algorithm and of **VT-5** were was determined from the structural evolution. The structures of **VT-1**, **Cs-VT-1**, and **VT-5** were refined by powder XRD Rietveld refinement using the Materials Studio package (Accelrys Software Inc.)[59]. The pattern parameters and lattice parameters of the materials were refined by the Pawley method. Then, isotropic temperature factors were given for every atom in the initial structure of the material. Rietveld analysis was initiated with the initial models of the materials, and the lattice parameters and pattern parameters were from the Pawley refinement. Every atom position was refined. The occupancy of the atoms in the framework was fixed without further refinement, and the occupancies of the atoms in water and the cations were refined with consideration of the elemental analysis results. Finally, the pattern parameters were refined again to obtain the lowest $R_{wp}$ value. The crystallographic parameters and atom positions of the materials are shown in Supplementary Table 4, Table 23, and Table 21.

**Computer-based calculations and simulations**. Structural modelling and DFT calculation were carried out on the Materials Studio package (Accelrys Software Inc.). For the DFT calculation, the structure of the material was optimized using the DMol$^3$ program[60,61]. We employed the Perdew–Burke–Ernzerhof (PBE) generalized gradient functional and DND basis set. After geometry optimization, the energy and density value were obtained. The porosity of the zeolitic **VTs** and the reported zeolites was calculated from the Eq. (1)

$$porosity = \frac{[accessible\ free\ space]}{[total\ space]} \quad (1)$$

The free space and total space of the material were calculated by the Connolly surface estimation using the function of the 'Atom Volumes & Surfaces' dialogue. The structures of the reported zeolites were available from the website[62]. Formation energy ($\Delta H_f$) calculation that predicted the stability of solid material. Generally, the formation energy is given by the Eq. (2)

$$\Delta H_f = E_{total} - \sum_i \mu_i x_i \quad (2)$$

where $E_{total}$ is the DFT total energy of the compound, $\mu_i$ is the chemical potential of element $i$ and $x_i$ is the number of element $i$ in the compound[63]. The initial structures of the molecular type **VTs** were obtained from previous study[28]. The crystalline water was eliminated to accord to the situation of zeolitic **VTs**. The material structures were optimized by DFT calculation.

**Structural evolution and prediction**. The difficulty in predicting structure of transition metal oxides compared with metal-organic frameworks and zeolites is that the chemical composition of transition metal oxides might change with changes in the assembly style, which causes the difficulty in comparing the energies of the hypothetical structures, because the materials might not be isomeric.

To build the library of isomeric structure for **VT** based on $[W_4O_{16}]^{8-}$ units with the $VO^{2+}$ linker, there were two restrictions for generating isomeric structure for energy comparison, minimizing the number of the hypothetical structures, and saving time in the theoretical calculation. (1) Each terminal W=O bond must connect to one and only one $VO^{2+}$ linker. (2) Each $VO^{2+}$ linker must connect to only two $[W_4O_{16}]^{8-}$ units by two bonds of each units.

The possibilities for the connections of the $[W_4O_{16}]^{8-}$ units with $VO^{2+}$ linkers were proposed. There were 12 terminal W=O bonds. Based on the restrictions, 6 linker ions were necessary. Four $WO_6$ octahedra were in the $[W_4O_{16}]^{8-}$ cluster. These octahedra are denoted as blue octahedron (b), grey octahedron (g), purple octahedron (p), and yellow octahedron (y) (Supplementary Fig. 8). There are two different connection styles. Type i was formed by connection of the linker ion with

a W=O terminal bond from two different $WO_6$ octahedra, and type ii was formed by connection of a linker ion to a single $WO_6$ octahedron (Supplementary Fig. 9).

For the type i connection, there were six possible octahedra combinations, (b, g), (b, p), (b, y), (g, p), (g, y), and (p, y). All the possible connections of the $[W_4O_{16}]^{8-}$ units with V linkers were listed in Supplementary Table 7. Only the cases in entries 1, 23, 30, 37, 41, 47, 50, and 52 were reasonable structures. Entry 1 was ascribed to preliminary building unit A, entries 23, 30, 37, and 41 were ascribed to preliminary preliminary building unit B, and entries 47, 50, 52 were ascribed to preliminary building unit C. These three preliminary building units further connected to form secondary building units. After the hypothetical structures were constructed, the symmetry of the frameworks was determined by the "find symmetry" function in the Materials Studio software package.

The crude structures were relaxed by DFT calculation using the DMol$^3$ code in the Materials Studio software package. The cell parameters were also optimized during geometry optimization for each hypothetical structure.

**Cation adsorption**. Ion-exchange. K$^+$ in **VT-1** were exchanged with CsCl in an aqueous solution. First, 0.1 g of **VT-1** was dispersed in a CsCl aqueous solution (0.2 g, 5 mL). The solution was stirred at room temperature for 1 h. The resulting ion-exchanged material was recovered by centrifugation (2330 × g, 5 min). The resulting material was washed with water (5 mL) 3 times. The material was dried at 60 °C overnight. Elemental Analysis: Calcd for $Cs_{1.75}W_4V_3O_{27}H_{16.25}$: N, 0; K, 0; Cs, 14.82; W, 46.87; V, 9.74; H, 1.04, Found: N, 0; K, 0; Cs, 14.71; W, 46.79; V, 9.98; H 0.97.

**Molecule adsorption**. Gas adsorption experiments were carried out on a BEL sorp max system (BEL, Japan). Before adsorption the materials were treated at 150 °C for 2 h under high vacuum. N$_2$ adsorption–desorption measurements were conducted at −196 °C. The surface area, pore volume, and pore size distribution were estimated by the Brunauer-Emmett-Teller (BET) method, t-plot method, and the SF method, respectively, using the adsorption blanch of the isotherms.

Various alkanes with different kinetic diameters were used as molecular probes, including methane (0.38 nm)[64], ethane (0.4 nm)[64], propane (0.43 nm)[65], $n$-butane (0.43 nm)[64], $i$-butane (0.5 nm)[66], and cyclohexane (0.60 nm)[67]. The adsorption measurements were carried out at 25 °C to evaluate the porosity of the material. The resulting adsorption isotherms were analysed using the DA[68] equation to obtain the pore volume.

$$W = W_0 \exp\left[\left(-\frac{A}{\beta E_0}\right)^m\right] \quad (3)$$

$W_0 = a_0 v$ is the limiting micropore volume, where $a_0$ is the limiting amount adsorbed, and $v$ is the molar volume of the adsorbate. In the Eq. (3), $A$ is Polanyi's adsorption potential, which is calculated from the saturated vapour pressure $p_0$ and the adsorption pressure $p$ using $A = RT\ln(p_0/p)$. $\beta$ is the similarity coefficient of the characteristic curves. $m$ is a structure-related parameter and $W$ is the adsorbed volume. $E_0$ is a characteristic energy that depends on the microporous structure of a given adsorbent $W_0$ can be determined from the intercept of the plot of $\ln W$ vs. $\ln^m(p_0/p)$ ($m = 3$) assuming that the adsorbed phase has the same density as the bulk liquid.

**NH$_3$-SCR**. NH$_3$-SCR was conducted over **VT-1** (0.15 g) using a fixed bed reactor at atmospheric pressure. V$_2$O$_5$, VO$_2$, and WO$_3$ were purchased from Wako (Japan), which were ball-milled for 100 min. XRD and TEM demonstrated that the structures of the materials did not change and the particle sizes were smaller than **VT-1** (Supplementary Fig. 24 and Fig. 25). **VT-1** and all the reference catalysts were sieved to control the secondary particle size to be 38–106 μm. Before the catalytic reaction, **VT-1** was treated at 150 °C under a 250 mL min$^{-1}$ flow of Ar for 1 h. After the heat treatment, the system was cooled to a set temperature and then the reaction gas was introduced at a flow rate of 250 mL min$^{-1}$. The composition of the gas consisted of NO (250 ppm), NH$_3$ (250 ppm), and O$_2$ (4 vol%) with Ar as a diluter. The catalytic reaction was carried out at set temperatures of 40, 60, 100, 120, 150, and 200 °C and the outlet gas was monitored for 120 min. The reactants and products were analysed with an FT-IR instrument (FT/IR-4700ST, JASCO) equipped with a gas cell (LPC-12M-S, light path length 12 m, JASCO). The conversion of NO was calculated by the Eq. (4).

$$Conv._{NO} = \frac{NO_{in} - NO_{out}}{NO_{in}} \times 100 \quad (4)$$

The conversion of NH$_3$ was the same level of that of NO. We confirmed that the concentrations of NO$_2$ and N$_2$O were below the detection limit (0.5 ppm), and therefore the selectivity for N$_2$ calculated by the Eq. (5) more than 99% for all experiments.

$$Sel._{N_2} = \frac{(NO_{in} + NH_{3\,in}) - (NO_{out} + NH_{3\,out} + NO_{2\,out} + N_2O_{out})}{(NO_{in} + NH_{3\,in}) - (NO_{out} + NH_{3\,out})} \times 100 \quad (5)$$

## Data availability

The authors declare that the data supporting the findings of this study are available within the main article and the Supplementary Information. Additional data are available from the corresponding authors on reasonable request.

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

## Acknowledgements

This work was supported by a Grants-in-Aid for Scientific Research (A) from the Ministry of Education, Culture, Sport, Science, and Technology of Japan (MEXT) (grant number: 15H02318). The authors also thank the Material Analysis Suzukake-dai Center, Technical Department, Tokyo Institute of Technology, for the elemental analysis.

## Author contributions

Z.Z. and W.U. designed the research. Z.Z. and Q.Z. performed the material synthesis, characterization, and other corresponding experiments. Z.Z. performed the structural evolution studies and structural determination. T.M., S.H., and Z.Z. carried out the NO reduction reactions. N.H. performed the HAADF-STEM observations. A.Y. and H.Y. conducted XAFS measurements. M.H. and S.I. offered experimental assistance. Z.Z. and M.S. wrote the paper.

## Additional information

**Competing interests:** The authors declare no competing interests.

