## [Peer Review File · Nature Communications]

PEER REVIEW FILE

Reviewers' comments:

Reviewer #1 (Remarks to the Author):

The authors have addressed all my concerns and substantially improved their initial submission. I am pleased to recommend now this manuscript for publication in Nature Comm. However, before publication two references should be added to the main text:

(1) in the text part, discussing Tschörtnerite and the [Cu₁₂O₂₄] cluster, to link to the chemistry of polyoxometalates: Chem. Eur. J. 2017, 23, 7841–7852.

(2) in the text part, indicating XPS in order to link the determination of reduced V ions to that in the chemistry of polyoxometalates: J. Phys. Chem. C 2017, 121, 10419–10429.

The quantitative data of the XPS analysis (O 1s, V 2p_{1/2}, V 2p_{3/2} core levels, etc.) should be briefly described in the main text.

Reviewer #2 (Remarks to the Author):

Obtaining new porous materials is a current endeavour of many groups, their result is important, but they should also mention the work of other researchers in the area, such as work regarding the most recent advances in POM framework materials: Vilà-Nadal, L. & Cronin, L. Design and synthesis of polyoxometalate-framework materials from cluster precursors. Nat. Rev. Mater. 2, 17054 (2017). In this case they use vanadotungstates that exhibit large porosity. This is a remarkable result in the area.

Novelty: Mo and Vanadium mixed metal oxide materials have been described by the same group in Crystalline Mo₃VO_x mixed metal oxide catalyst with trigonal symmetry. Angew. Chem., Int. Ed., 2007, 46, 1493–1496.

The formation of porous materials with tungsten is expected. Still in my opinion their results show a new family of stable framework materials which merits publication.

The authors did a good job characterizing the compounds and describing its adsorbing and catalytic activity. They also performed predictive theoretical calculations based on the work of Pulido, A. et al. Functional materials discovery using energy-structure-function maps. Nature 543, 657–664 (2017). Their results are reasonable and well explained within the text.

Overall this is good work, and I recommend publication in Nature Comms subject to the following:

1) mention design in context as explained by Vilà-Nadal, L. & Cronin, L. Design and synthesis of polyoxometalate-framework materials from cluster precursors. Nat. Rev. Mater. 2, 17054 (2017).

2) Can the authors comment on their catalytic activity vs current commercial available catalysts?

A comparison with other selective catalytic materials available would certainly help to calibrate the work. NH₃-SCR is a hot topic (over 200 papers in 2017) and there are lots of very good catalyst available.

3) The authors should make at least a mention to other available catalysts in context.

Reviewer #3 (Remarks to the Author):

I see a thorough demand of revision from the other referees as well but I will concentrate only on the answers to my criticism : My demands regarding the measurements of the catalytic activity displayed in Fig 6 have been met.

Now a more deep discussion is needed regarding e.g. the maximum of VT-1 found around 130 C while the more conventional catalysts have no maximum. These types of catalysts have usually a maximum about 350 - 450 C interpreted as commencing oxidation of ammonia, thus the catalysts are oxidizing catalysts after the maximum temperature but in fact also all ready of importance below the maximum. My question is therefore if SO₂ even to a small extent is oxidized by VT-1 at 130 C ? For industrial use this is critical since even 10-20 ppm of SO₃ possibly formed is unacceptable in the stack emission.

It seems that the authors are unwilling to inform about this because - as they claim- the company performing the measurement want to keep the information secret !

It should anyway be noted that even in dry gas the VT-1 catalyst performs inferior to the traditional catalysts at temperatures above 160 C (Fig 6(b)). Then knowing from the revised manuscript that water is severely deactivating the catalyst at low temperatures (Fig R5) which is known not be the case for conventional mesoporous catalysts, there is no hope of commercial demand of the VT-1 type of catalyst unless the flue gas is completely dry and may be without even small amounts of SO₂. These conditions are essentially only of academic interest.

In this connection it should be noted that the water content measured of 2.2 % corresponds to saturation of the gas at room temperature. Most industrial flue gases have a water content of 7-15 % H₂O. With the pore size of VT-1 of around 4 Å I am convinced that the pores will be flooded by condensed water even at temperatures well above 150 C. I suggest that the authors employ the Kelvin equation to estimate the pore condensation temperatures for the water concentration range 2- 15 % if they do not actually measure it experimentally.

I am satisfied that the authors admit the comparison in the old table S24 as being absurd and consequently omit this table.

I consider my needed revisions as major because of the serious character of my inquiries. The answers should be included in the text and in the Conclusion as well.

Reviewers' comments:

Reviewer #1 (Remarks to the Author):

The authors have addressed all my concerns and substantially improved their initial submission. I am pleased to recommend now this manuscript for publication in Nature Comm. However, before publication two references should be added to the main text:

(1) in the text part, discussing Tschörtnerite and the [Cu₁₂O₂₄] cluster, to link to the chemistry of polyoxometalates: Chem. Eur. J. 2017, 23, 7841–7852.

Our response: Thank you very much for your suggestion. The paper for the [Cu₁₂O₂₄] cluster was cited in the revised manuscript. The corresponding part was modified. Please check page 5, paragraph 3.

(2) in the text part, indicating XPS in order to link the determination of reduced V ions to that in the chemistry of polyoxometalates: J. Phys. Chem. C 2017, 121, 10419–10429.

Our response: Thank you very much for your suggestion. The paper for XPS was mentioned and cited in the revised manuscript. Please check page 4, paragraph 4.

The quantitative data of the XPS analysis (O 1s, V 2p_{1/2}, V 2p_{3/2} core levels, etc.) should be briefly described in the main text.

Our response: Thank you very much for your suggestion. XPS was conducted for O(1s), showing two different simulated peaks, which indicated the existence of a framework O with low binding energy and surface adsorbates with high binding energy (Figure R 1).¹ The ratio of O(surface)/O/framework) was 0.67 and 1.47 for **VT-1** and **VT-5**, respectively. For V, V (2p_{3/2}) was more typical, because V (2p_{1/2}) peak was broad and did not overlap with V (2p_{3/2}) peak. W was all W^{VI} (quantitatively 100%) and V was all V^{IV} (quantitatively

100%), which were confirmed by XPS and XAFS. The responding parts were modified.
Please check page 4, paragraph 4.

Figure R 1. XPS profiles. (a) W in VT-1, (b) V in VT-1, (c) O in VT-1, (d) W in VT-5, (e) V in VT-5, and (f) O in VT-5

Reviewer #2 (Remarks to the Author):

Obtaining new porous materials is a current endeavour of many groups, their result is important, but they should also mention the work of other researchers in the area, such as work regarding the most recent advances in POM framework materials: Vilà-Nadal, L. & Cronin, L. Design and synthesis of polyoxometalate-framework materials from cluster precursors. *Nat. Rev. Mater.* 2, 17054 (2017). In this case they use vanadotungstates that exhibit large porosity. This is a remarkable result in the area.

Novelty: Mo and Vanadium mixed metal oxide materials have been described by the same group in Crystalline Mo₃VO_x mixed metal oxide catalyst with trigonal symmetry. *Angew. Chem., Int. Ed.*, 2007, 46, 1493–1496.

The formation of porous materials with tungsten is expected. Still in my opinion their

results show a new family of stable framework materials which merits publication.

The authors did a good job characterizing the compounds and describing its adsorbing and catalytic activity. They also performed predictive theoretical calculations based on the work of Pulido, A. et al. Functional materials discovery using energy-structure-function maps. Nature 543, 657–664 (2017). Their results are reasonable and well explained within the text.

Overall this is good work, and I recommend publication in Nature Comms subject to the following:

1) mention design in context as explained by Vilà-Nadal, L. & Cronin, L. Design and synthesis of polyoxometalate-framework materials from cluster precursors. Nat. Rev. Mater. 2, 17054 (2017).

Our response: Thank you very much for your suggestion. The design from the paper (Vilà-Nadal, L. & Cronin, L. Design and synthesis of polyoxometalate-framework materials from cluster precursors.) was mentioned in the introduction and the paper was cited in the revised manuscript. Please check page 3, paragraph 1.

2) Can the authors comment on their catalytic activity vs current commercial available catalysts?

A comparison with other selective catalytic materials available would certainly help to calibrate the work. NH₃-SCR is a hot topic (over 200 papers in 2017) and there are lots of very good catalyst available.

Our response: Thank you very much for your suggestion. V₂O₅-WO₃/TiO₂ type catalysts are widely used commercial available catalysts,^{2,3} and we synthesized V₂O₅-WO₃/TiO₂ for comparison with **VT-1** under the same reaction conditions. Furthermore, we also synthesized V₂O₅/TiO₂ catalyst for comparison. The results showed that **VT-1** was more

active than $V_2O_5-WO_3/TiO_2$ and V_2O_5/TiO_2 (Figure 6b) when the temperature was below 160 °C.

We agree that comparison of **VT-1** with other available catalytic materials is important to understand the activity of **VT-1**. However, as the reviewer3 suggested in the last peer review comments (please check the last peer review comment of reviewer3 at the end of this file) that we could not compare the activity from the paper directly because the reaction conditions were not exactly the same. Therefore, we synthesized two catalysts, $V_2O_5-WO_3/TiO_2$ and V_2O_5/TiO_2 , for comparison with our catalyst under the same conditions, because $V_2O_5-WO_3/TiO_2$ and V_2O_5/TiO_2 were typical (having similar composition to **VT-1**) and widely used.

This part has been mentioned in main text. Please check page 10, paragraph 3.

3) The authors should make at least a mention to other available catalysts in context.

Our response: Thank you very much for your suggestion. We mentioned and cited other catalysts, such as V-based catalysts,⁴⁻⁹ Mn-based catalysts,¹⁰ and Cu-zeolite catalysts¹¹ in the context. Please check page 10, paragraph 1.

Reviewer #3 (Remarks to the Author):

I see a thorough demand of revision from the other referees as well but I will concentrate only on the answers to my criticism : My demands regarding the measurements of the catalytic activity displayed in Fig 6 have been met.

Now a more deep discussion is needed regarding e.g. the maximum of VT-1 found around 130 C while the more conventional catalysts have no maximum. These types of catalysts have usually a maximum about 350 - 450 C interpreted as commencing oxidation of ammonia, thus the catalyts are oxidizing catalyts after the maximum temperature but in fact also all ready of importance below the maximum.

Our response: Thank you very much. The maximum of **VT-1** was different from that of the conventional catalysts. The maximum of **VT-1** was due to the unstable framework at a high reaction temperature (200 °C) with O₂, which we have demonstrated in the main text (supplementary Figure 19).

My question is therefore if SO₂ even to a small extent is oxidized by VT-1 at 130 C ? For industrial use this is critical since even 10-20 ppm of SO₃ possibly formed is unacceptable in the stack emission.

It seems that the authors are unwilling to inform about this because - as they claim- the company performing the measurement want to keep the information secret !

Our response: Thank you very much for your question. We did not have special instrument to control and analyze SO₂ and SO₃. Therefore, we are very sorry that we cannot answer the reviewer's question directly.

We tried our best to response the reviewer's question in an indirect way. We assembled a simple reactor to generate SO₂ that further reacted with **VT-1** at 130 °C for 1 h. HCl (37%) was dropped (4 drops per minute in average) to Na₂SO₃ (20 g) continuously to form SO₂ gas that was flowed to **VT-1** at 130 °C directly for 1 h. After the treatment, **VT-1** was characterized by XRD to make sure that the structure did not change (Figure R 2a,b). XPS was used for investigating the oxidation states of the elements. Compared with the as-synthesized **VT-1**, the oxidation states of W and V were reduced after the treatment, which might indicate that SO₂ oxidized **VT-1** at 130 °C. In this experiment, the amount of SO₂ would be large. We did not know the result when the content of SO₂ is small.

The information was added in the revised manuscript. Please check page 11, paragraph 1.

Figure R 2. Characterizations of **VT-1** after SO_2 treatment. XRD patterns of **VT-1** (a) before and (b) after SO_2 treatment at $130\text{ }^\circ\text{C}$, XPS profile of W of **VT-1** (c) before and (d) after SO_2 treatment at $130\text{ }^\circ\text{C}$, XPS profile of V of **VT-1** (e) before and (f) after SO_2 treatment at $130\text{ }^\circ\text{C}$.

It should anyway be noted that even in dry gas the VT-1 catalyst performs inferior to the traditional catalysts at temperatures above $160\text{ }^\circ\text{C}$ (Fig 6(b)). Then knowing from the revised manuscript that water is severely deactivating the catalyst at low temperatures (Fig R5) which is known not be the case for conventional mesoporous catalysts, there is no hope of commercial demand of the VT-1 type of catalyst unless the flue gas is completely dry and may be without even small amounts of SO_2 . These conditions are essentially only of academic interest.

Figure R 3. XRD patterns of (a) **VT-1**, (b) **VT-1** calcined at $200\text{ }^\circ\text{C}$ in air, (c) Ti exchanged **VT-1**, (d) Ti exchanged **VT-1** calcined at $200\text{ }^\circ\text{C}$ in air, (e) Mn exchanged **VT-1**, and (f) Mn exchanged **VT-1** calcined at $200\text{ }^\circ\text{C}$ in air

Our response: Thank you very much for your comments. The poor activity of **VT-1** above $160\text{ }^\circ\text{C}$ resulted from low stability of the material in O_2 at a high temperature (supplementary Figure 19). The material was calcined at $200\text{ }^\circ\text{C}$ in air, and XRD showed

that the structure collapsed completely (Figure R 3a,b).

We agree with the reviewer's statement that in the view of industrial application the performance of **VT-1** at a high temperature (200 °C) and in the presence of water was not good. There is a long way to go to solve the issues and commercialize **VT-1**.

However, the purpose of this paper is not commercialization of **VT-1** for practical application in NH₃-SCR. The primary purpose of this manuscript is to show the creation of the new family of microporous **VTs**, the detailed structural characterizations, and the property investigations. Owing to complexing acidity, redox property, and microporosity in **VT-1**, we selected NH₃-SCR as a model reaction to test the catalytic activity, because high performance NH₃-SCR catalysts needed acidity, redox property, and high surface area (porosity). The reaction conditions we used were ideal (dry gas without SO₂), but under the same reaction conditions **VT-1** was more active at a low temperature (120 °C) than conventional catalysts (V₂O₅-WO₃/TiO₂ and V₂O₅/TiO₂) (Figure 6b), which confirmed the superiority of the complexity, and it is very interesting at least in the fundamental catalysis researches.

For practical application, we are still positive. Currently, the problems of **VT-1** for NH₃-SCR, such as low activity at a high temperature and with water, are true. However, the composition and structure of the **VT** materials are highly tunable, which have been demonstrated in the main text. There are chances to improve the performance by modification of the composition or structure of **VTs**.

For example, the stability of **VT-1** in air at a high temperature was able to be improved by introducing transition metal ions by ion-exchange. After introducing Mn or Ti in **VT-1**, the basic structure of **VT-1** did not change while the stability of the material in air at 200 °C increased remarkably (Figure R 3c-f), which is promising. However, this research based ion-exchange should be deeply studied, such as the position of Mn or Ti, the local structure of Mn or Ti species in **VT-1**, the mechanism of the stabilization of the framework, and the catalytic activity (with H₂O or SO₂), and these investigations are still on going and cannot be finished in a short period, which would be our next step. All the further researches on **VTs** are based on the material synthesis, the structural characterizations, and the property investigations described in this manuscript. Therefore, we would like to

publish this important manuscript first.

In this connection it should be noted that the water content measured of 2.2 % corresponds to saturation of the gas at room temperature. Most industrial flue gases have a water content of 7-15 % H₂O. With the pore size of VT-1 of around 4 Å I am convinced that the pores will be flooded by condensed water even at temperatures well above 150 C. I suggest that the authors employ the Kelvin equation to estimate the pore condensation temperatures for the water concentration range 2- 15 % if they do not actually measure it experimentally.

Our response: Thank you very much for your suggestion. We applied the Kelvin equation¹² to estimate physical adsorption of water and NH₃ and their competition.

$$\ln p/p_0 = -2\sigma M/(RT\rho r)$$

In this equation, p is the vapour pressure of the adsorbate for capillary condensation (pore filling). p_0 is the saturated pressure of the adsorbate. σ is the surface tension of the adsorbate. M is the molar mass of the adsorbate. R is the ideal gas constant. T is the temperature. ρ is the density of the adsorbate, r is radius of curvature when pore filling occurs.

R and M are constants. σ , ρ , and p_0 are temperature dependent values, which are obtained or estimated according to the literatures at different temperatures (Table R 1 and Table R 2). r is determined using the experimental water adsorption isotherm (Figure 5e in main text) at 25 °C by the Kelvin equation and proposed to be constant for water and NH₃. $r = 3.32710 \times 10^{-10}$ m.

After having the above information, we estimated the vapour pressure (p) for capillary condensation (pore filling) at different temperatures. When the water content is 11.77%-18.47%, the condensation temperature in the micropore of **VT-1** is 100-110 °C (Table R 1). When the water content is 1.47%-2.60%, the condensation temperature is 60-70 °C. For NH₃, condensation is more difficult than water. The vapour pressure (p) for

capillary condensation (pore filling) of NH₃ is higher than that of water. Therefore, water preferentially adsorbs in **VT-1**, and water would inhibit the reaction of **VT-1**.

The Kelvin equation estimation is only based on physical adsorption. NH₃ also has chemical interaction with **VT-1**. NH₃-TPD clearly shows that **VT-1** adsorbs NH₃ at 100 °C, corresponding to the chemical adsorption (supplementary Figure 20). Chemical adsorption is usually slower but stronger than physical adsorption. Therefore, water is likely to be adsorbed in **VT-1** rapidly, but still a part of NH₃ can be adsorbed and reacted with NO. The explanation has been added in the revised manuscript. Please check page 11, paragraph 1.

Table R 1. The relationship between the temperature and the vapour pressure of water when capillary condensation (pore filling) occurred estimated by the Kelvin equation.

Temperature (°C)	ρ (kg m ⁻³) ¹³	σ (N m ⁻¹) ¹⁴	ρ_0 (kPa) ¹⁵	p (kPa)	Vol% ^a
150	916.1287819	0.04874	472.5519086	92.01	90.80
140	925.3326778	0.05085	358.9651729	63.57	62.74
130	934.0911806	0.05293	268.706703	43.14	42.57
120	942.4097092	0.05496	197.9717544	28.72	28.34
110	950.2885172	0.05696	143.3647696	18.72	18.47
100	957.7223216	0.05891	101.8929744	11.92	11.77
90	964.6997829	0.06082	70.95431039	7.41	7.31
80	971.2027609	0.06267	48.3199255	4.48	4.42
70	977.2052245	0.06447	32.11164611	2.63	2.60
60	982.6716125	0.06624	20.77505759	1.49	1.47
50	987.5542946	0.06794	13.04901599	0.82	0.81
40	991.7894952	0.0696	7.932584148	0.43	0.42
30	995.2904755	0.0712	4.650516236	0.22	0.21

^a volume percentage in 101.325 kPa.

Table R 2. The relationship between the temperature and the vapour pressure of NH₃ when capillary condensation (pore filling) occurred estimated by the Kelvin equation.

Temperature (°C)	ρ (kg m ⁻³) ¹⁶	σ (N m ⁻¹) ^{17, a}	ρ_0 (kPa) ^{18, b}	p (kPa)	Vol%
150	485.8	-	15067.64774	-	-
140	497.5	0.00129	12821.25888	11869.4	-
130	509.8	0.00322	10840.44541	8941.6	-
120	522.8	0.00452	9101.395264	6945.7	-
110	536.4	0.00548	7582.092755	5463.3	-
100	550.9	0.00709	6262.159055	4098.4	-
90	566.2	0.0071	5122.695184	3351.0	-

80	582.4	0.00871	4146.130839	2463.7	-
70	599.7	0.01064	3316.082462	1756.4	-
60	618.2	0.01258	2617.224105	1235.3	-
50	638	0.01484	2035.17472	840.2	-
40	659.3	0.01677	1556.405529	573.5	-
30	682.2	0.01903	1168.170854	377.0	-

^a The data is from theoretical equation at 100-140 °C. ^b The data is out of range of the equation at 80-150 °C.

I am satisfied that the authors admit the comparison in the old table S24 as being absurd and consequently omit this table.

I consider my needed revisions as major because of the serious character of my inquiries. The answers should be included in the text and in the Conclusion as well.

Our response: Thank you very much for your suggestion. We mentioned the problems that **VT-1** suffered currently in the revised manuscript. Please check page 11, paragraph 1 and 2.

In summary, we agree with reviewer3's criticism on the catalytic performance of **VT-1** in the view of industrial applications, such as the unstable structure at a high temperature and water inhibition. All of these would limit the practical applications of **VT-1** to NH₃-SCR. Therefore, further modification of **VT-1** to enhance the stability, water durability, and SO₂ durability for NH₃-SCR is necessary.

However, this paper does not mainly focus on the practical applications of **VT-1** but focuses on the interesting structure and properties of the material. Fortunately, due to the unique structure, **VT-1** shows superior activity to the conventional catalysts under ideal reaction conditions of NH₃-SCR at low temperature. Furthermore, **VTs** have high structure and composition tunability, which enables the further modification of the materials and might overcome the above problems in the future. All the further researches are still on going and based on the study described in this manuscript. Therefore, we would like to report the synthesis, characterization, and property investigation of the material first.

References

1. O. Linnenberg, M. Moors, A. Solé-Daura, X. López, C. Bäumer, E. Kentzinger, W. Pyckhout-Hintzen, K. Y. Monakhov. "Molecular characteristics of a mixed-valence polyoxovanadate $\{V^{IV/V}_{18}O_{42}\}$ in solution and at the liquid–surface interface". *J. Phys. Chem. C* **121**: 10419–10429 (2017).
2. C. Chen, W. Jia, S. Liu, Y. Cao. "Simultaneous NO removal and Hg⁰ oxidation over CuO doped V²O⁵-WO³/TiO² catalysts in simulated coal-fired flue gas". *Energy & Fuels* **32**: 7025–7034 (2018).
3. L. Ji, X. Cao, S. Lu, C. Du, X. Li, T. Chen, A. Buekens, J. Yan. "Catalytic oxidation of PCDD/F on a V₂O₅-WO₃/TiO₂ catalyst: effect of chlorinated benzenes and chlorinated phenols". *J. Hazard. Mater.* **342**: 220–230 (2018).
4. Y. He, M. E. Ford, M. Zhu, Q. Liu, U. Tumuluri, Z. Wu, I. E. Wachs. "Influence of catalyst synthesis method on selective catalytic reduction (SCR) of NO by NH₃ with V₂O₅-WO₃/TiO₂ catalysts". *Appl. Catal. B Environ.* **193**: 141–150 (2016).
5. T. Boningari, R. Koirala, P. G. Smirniotis. "Low-temperature catalytic reduction of NO by NH₃ over vanadia-based nanoparticles prepared by flame-assisted spray pyrolysis: Influence of various supports". *Appl. Catal. B Environ.* **140-141**: 289–298 (2013).
6. I. Song, S. Youn, H. Lee, S. G. Lee, S. J. Cho, D. H. Kim. "Effects of microporous TiO₂ support on the catalytic and structural properties of V₂O₅/microporous TiO₂ for the selective catalytic reduction of NO by NH₃". *Appl. Catal. B Environ.* **210**: 421–431 (2017).
7. W. Cha, S. Chin, E. Park, S. Yun, J. Jurng. "Effect of V₂O₅ loading of V₂O₅/TiO₂ catalysts prepared via CVC and impregnation methods on NO_x removal". *Appl. Catal. B Environ.* **140-141**: 708–715 (2013).
8. B. Liu, J. Du, X. Lv, Y. Qiu, C. Tao. "Washcoating of cordierite honeycomb with vanadia-tungsta-titania mixed oxides for selective catalytic reduction of NO with NH₃". *Catal. Sci. Technol.* **5**: 1241–1250 (2015).
9. P. G. W. A. Kompio, A. Brückner, F. Hipler, G. Auer, E. Löffler, W. Grünert. "A new view on the relations between tungsten and vanadium in V₂O₅-WO₃/TiO₂ catalysts for the selective reduction of NO with NH₃". *J. Catal.* **286**: 237–247 (2012).
10. S. Zhan, M. Qiu, S. Yang, D. Zhu, H. Yu, Y. Li. "Facile preparation of MnO₂ doped Fe₂O₃ hollow nanofibers for low temperature SCR of NO with NH₃". *J. Mater. Chem. A* **2**: 20486–20493 (2014).

11. F. Gao, D. Mei, Y. Wang, J. Szanyi, C. H. F. Peden, F. Gao, D. Mei, Y. Wang, J. Szanyi, C. H. F. Peden. "Selective catalytic reduction over Cu/SSZ-13: linking homo- and heterogeneous catalysis". *J. Am. Chem. Soc.* **139**: 4935–4942 (2017).
12. G. Yin, Q. Liu, Z. Liu, W. Wu. "Extension of Kelvin equation to CO₂ adsorption in activated carbon". *Fuel Process. Technol.* **174**: 118–122 (2018).
13. F. E. Jones, G. L. Harris. "ITS-90 Density of water formulation for volumetric standards calibration". *J. Res. Natl. Inst. Stand. Technol.* **97**: 335–340 (1992).
14. N. B. Vargaftik, B. N. Volkov, L. D. Voljak. "International Tables of the Surface Tension of Water". *J. Phys. Chem. Ref. Data* **12**: 817–820 (1983).
15. C. Antoine. "Tensions des vapeurs: nouvelle relation entre les tensions et les temperatures". *Compt. Rend. Acad. Sci.* **107**: 681–684, 778–780, 836–837. (1888).
16. "Engineering ToolBox, (2018). Ammonia-density at varying temperature and pressure. [online] Available at: https://www.engineeringtoolbox.com/ammonia-density-temperature-pressure-d_2006.html".
17. G. J. Gloor, G. Jackson, F. J. Blas, E. Marti, E. De Miguel, G. J. Gloor, G. Jackson, E. De Miguel. "An accurate density functional theory for the vapor-liquid interface of associating chain molecules based on the statistical associating fluid theory for potentials of variable range". *J. Chem. Phys.* **121**: 12740–12759 (2004).
18. C. S. Cragoe, C. H. Meyers, C. S. Taylor. "The vapor pressure of ammonia". *J. Am. Chem. Soc.* **42**: 206–229 (1920).

----- reviewer3's comment of last peer review-----

Reviewer #3 (Remarks to the Author):

The paper is dealing with an interesting subject as transition metal zeolites synthesis and characterization.

I am not a crystallographer so I am not able to judge fully if the powder data are sufficient to draw all conclusions about the detailed structure described in the paper.

However, regarding the catalytic part I have several concerns :

The reported SCR activity of the VT-1 catalyst lacks information on the particle size

fraction investigated since a simple powder sample will lead to an uncontrolled measured activity impossible to compare with work in the literature.

Therefore the comparison in Figure 6 is not justified unless the samples are run with the same size fraction and also referred to the same sample mass or molar amount of active metal.

Furthermore the comparison in table S.24 is absurd unless all referred samples are run under the same conditions regarding :

Amount and size fraction, flow of gas, composition of the gas (NO, O₂, N₂ and e.g. with or without water, SO₂ etc ?).

Therefore the conclusion on p.9 that VT-1 is more active than those found in the literature is absurd. I guess one cannot even compare the published catalysts listed in the table .

Furthermore the catalyst prepared should be investigated in wet gas with SO₂ as well if relevant for most industrial applications including power and waste incineration plants.

This part of the paper needs major revision.